# Investigating the Adsorption Behavior of Polyaniline and Its Clay Nanocomposite towards Ammonia Gas

**DOI:** 10.3390/polym14214533

**Published:** 2022-10-26

**Authors:** Ahmed H. El-Shazly, Marwa Elkady, Amira Abdelraheem

**Affiliations:** 1Chemical and Petrochemicals Engineering Department, Egypt-Japan University of Science and Technology, Alexandria 21934, Egypt; 2Fabrication Technology Research Department, Advanced Technology and New Materials Research Institute (ATNMRI), City of Scientific Research and Technological Applications (SRTA-City), Alexandria 21934, Egypt; 3Environmental Health Department, High Institute of Public Health, Alexandria University, Alexandria 21524, Egypt

**Keywords:** polyaniline nanocomposites, ammonia adsorption, air pollution control

## Abstract

Air pollution and control of gaseous air pollutants are global concerns. Exposure to these gaseous contaminants causes several health risks, especially exposure to irritant gases such as ammonia (NH_3_). Furthermore, the application of smart polymeric nanocomposites in environmental applications has gained significant interest in recent years. In this study, aniline was polymerized without and with clay using a carbon dioxide (CO_2_)-assisted polymerization technique, yielding PANI and PANC samples, respectively. The samples were characterized using different methods, such as Fourier transform infrared spectroscopy (FTIR), X-ray diffraction (XRD), transmission electron microscopy (TEM), scanning electron microscope (SEM), and Brunner Emmett Teller (BET). The synthesized nanomaterials were utilized as gas adsorbents using a fixed bed reactor to investigate their adsorption behavior towards NH_3_. Three inlet NH_3_ concentrations were tested (35–150 ppm). The results revealed that the adsorption capacities of PANC nanocomposites were higher than nanostructured PANI for the studied concentrations. The adsorption capacities were 61.34 mgNH_3_/gm for PANC and 73.63 mgNH_3_/gm for PANI at the same inlet concentration (35 ppm). The highest NH_3_ adsorption capacity recorded was 582.4 mg NH_3_/gm, for PANC. This study showed the impressive adsorption behavior of the prepared PANI and PANC nanomaterials towards NH_3_ gas. Consequently, nanostructured PANI and PANC can be promising adsorbents that can be utilized to control different gaseous air pollutants.

## 1. Introduction

Adsorption is an efficient method that is commonly used either in air and/or water purification. The adsorption process has many advantages and has been widely experimented with because of high efficiency and its low cost, and the process is considered environment-friendly. Recently, ammonia (NH_3_) gas has gained a crucial role in industrial applications. However, it has serious health effects on exposed workers. The threshold limit value is 25 ppm, and the short-term exposure limit is 35 ppm for 10 min. The acute exposure to concentrations above these limits may cause irritation to the skin, nose, and eyes. Exposure to these concentrations for a longer time can cause severe illness. Therefore, adsorbents that have high NH_3_ selectivity and capacity are urgently desired [1]. Moreover, high concentrations of NH_3_ emissions may cause environmental pollution, affect the air quality, and intensify the aerosol particles and fog formation over industrial cities [2].

Polyaniline (PANI) and its nanocomposites have been extensively used as an adsorbent for removing contaminants, especially from aqueous solutions. In previous studies, the adsorption capacity of different PANI adsorbents was assessed towards water and wastewater contaminants. It was noticed that the adsorption capacities varied depending on several factors, mainly the preparation techniques. Recently, PANI and its nanocomposites have been synthesized through several methods and then used as adsorbents for the adsorption of aqueous contaminants, such as Mordant Black 11 dye, methyl red dye, sodium salicylate and metronidazole, and drug contaminants [3,4,5,6]. Additionally, its electrical and optical properties largely contributed to its utilization in gas sensing applications. PANI and its nanocomposites have been used for nitrogen dioxide (NO_2_), ammonia (NH_3_), water vapor (H_2_O), toluene (C_6_H_5_-CH_3_), methane (CH_4_), and carbon dioxide (CO_2_) gas sensing [7]. Alternatively, there were insufficient studies investigating the use of PANI or its composites as a gas adsorbent and/or an air filter. In 2019, polyaniline nanotubes have been prepared and experimented to remove NH_3_ and C_6_H_5_-CH_3_ from the air using a fixed bed reactor. Although the adsorption parameters were limited, the nanotubular structure of PANI showed high removal efficiency for the studied gases [8]. Charlotte Park et al. (2020) analyzed the physical filtration efficiency of PANI hybrid composite filter with graphite oxide for particulate matter 2.5. the hybrid filter composite showed a remarkable filtration PM_2.5_ efficiency, reaching 99.7 ± 0.08% [9]. Activated carbon/PANI nanocomposite has been used for CO_2_ gas removal using volumetric apparatus. The results indicated that the adsorption capacity of activated carbon was remarkably enhanced after adding PANI, because of the stronger affinity between CO_2_ and amine groups [10,11]. PANI-clay (PANC) composite and nanocomposite have been synthesized and characterized comprehensively in the literature. Clays are natural substances with interesting properties due to their geometries, surface area, and electrostatic charge. Incorporation of guest electroactive polymers like PANI into host clay particles has attracted great attention because of their better processability, alongside their colloidal stability, mechanical strength, and novel electrical and catalytic properties [12]. The main PANC applications were corrosion protection of steel surfaces and the removal of different aqueous contaminants [13,14,15,16]. Various clay-based composites have been prepared and utilized for gas adsorption. Copper-complexed clay/poly-acrylic acid composite was tested as NH_3_ gas adsorbent, and the highest adsorption capacity for NH_3_ was achieved by adding 75% of the clay, with capacities of 65.8 mg/g [17]. According to the best of our knowledge, the PANC nanocomposite was not used for NH_3_ adsorption. Various adsorbents were used in the adsorption of NH_3_. In 2021, the adsorption of NH_3_, starring with initial concentration 1700 ppm, by commercial zeolites to below 0.1 ppm have been examined, and the reported adsorption capacity reached 9.27 wt.% [18]. Huyen Thanh Vo et al. synthesized mesoporous alumina with controlled pore structure and used it in NH_3_ adsorption. It was reported that the adsorption capacity reached 193.57 mg/g [19]. Earlier, activated carbon has been utilized to treat NH_3_ emissions, and the adsorption capacity varied from 0.6 to 1.8 mg NH_3_/g carbon at 40 °C for inlet concentration ranged (600–2400 ppm) [20]. Ordered mesoporous carbon was synthesized by a self-assembly technique and utilized in NH_3_ adsorption. The equilibrium capacities at 298 K and 800 Torr was found to be 6.39 mmol/g [21]. A new type of zirconium-based metal organic framework has been prepared, characterized, and utilized in NH3 adsorption. It was stated that the adsorption capacity was recorded at 178.3 mg/g [22].

In this study, PANI and PANC were synthesized using the CO_2_-assisted polymerization technique. The prepared adsorbents were characterized using several characterization techniques, such as Fourier transform infrared spectroscopy (FTIR), X-ray diffraction (XRD), transmission electron microscopy (TEM), scanning electron microscope (SEM), and Brunner Emmett Teller (BET). The synthesized nanostructured materials were tested as adsorbents to remove both NH_3_ from simulated polluted air streams.

## 2. Materials and Methods

### 2.1. PANI and PANC Synthesis

Aniline, potassium persulphate (KPS), hydrochloric acid (HCl), methanol, clay, and toluene solution were purchased from Sigma-Aldrich (Darmstadt, Germany) and used as received. NH_3_ and CO_2_ gas cylinders with a purity of 99% were used as feed gas and mixed with nitrogen (N_2_) for preparing the required NH_3_ gas concentrations and used in CO_2_-assisted polymerization, respectively. Gas cylinders were purchased from Air supply (Alexandria, Egypt). All chemicals were of analytical grade, and solutions were prepared with freshly distilled water. Gas sensor (Drager Polytrone 5000) was purchased from Drägerwerk AG & Co. KGaA (Lübeck, Germany).

Nanostructured PANI was synthesized using the same technique described in our previous works [8,23,24]. The reactants were mixed inside a high-pressure vessel that is connected to a CO_2_ pump. CO_2_ was pressurized into the reactor at 10 MPa operating pressures and 40°C temperature for 3 h. After termination of the polymerization process, the produced precipitate was washed several times using hydrochloric acid, methanol, and distilled water, and then filtered by centrifugation and vacuum dried at 60 °C for 24 h. PANI was characterized using the following characterization techniques: Fourier transform infrared spectroscopy (FTIR), X-ray diffraction (XRD), transmission electron microscopy (TEM), scanning electron microscope (SEM), and Brunner Emmett Teller (BET).

For the PANC nanocomposite, the yield of the PANI samples was calculated, and 10% of its weight was equal to the added clay weight. The calculated clay weight was added before the polymerization, and the nanocomposite samples were then prepared using the same aforementioned PANI polymerization technique.

### 2.2. Gas Adsorption Experimental Setup

Standard NH_3_ concentration was prepared by mixing it with nitrogen (N_2_) gas. Both gases were passed through a mass flow controller and mixed homogeneously using several Y-connection consecutively, and then passed through long tubing to be measured using an NH_3_ (Drager Polytrone 5000) sensor [25]. Different NH_3_/N_2_ concentrations were prepared in the range of 35–150 ppm, as seen in Figure 1. The adsorption experiments were performed by entering the feed gas mixtures into a PANI and PANC nanocomposite fixed bed reactor. The bed consists of a glass tube with a dimension of 8 cm in length and 0.7 cm internal diameter. The gas adsorption experiments were attained until C/C_0_ = 0.90. The adsorption capacity of PANI and PANC nanocomposite towards NH_3_ was assessed by integrating the area under the breakthrough curve, as seen in Figure 1 [26].

## 3. Results

### 3.1. PANI/PANC Synthesis and Characterization

#### 3.1.1. Morphology Characterization

Harmonized nanorod PANI structure was prepared by using the SCCO_2_-assisted polymerization technique, as shown in Figure 2A,B. The use of SCCO_2_ facilitates the dissolving of reaction solutes together, owing to its high solvation power, besides producing a high-purity product. The high solvation power of SCCO_2_ enhances rapid polymerization reaction and prevents agglomeration of the produced nanostructure [27]. The mechanism of PANI nanorod formation, using different synthesis methods, was discussed by Haibing et al. [28]. They claimed that, at the initial stage, nanoparticles of PANI were created; these particles were then attached by a hydrogen bond, forming a rod-like structure. In our case, the presence of SCCO_2_ improved the linear growth of the nanostructure, resulting in fabricating uniform PANI-NR with an average diameter of approximately 80 nm.

The morphology of the synthesized PANC is shown in Figure 2C,D. Basically, in previous work, there are three types of composites that can be produced from the combination of PANI and clay. The immiscible structure of this nanocomposite includes the dispersion of clay aggregates within the polyaniline nanostructures. The polymer does not come in the clay layers; this type was obtained by different synthesis techniques. The second composite type is the intercalated structure that includes the formation of polyaniline chains between the clay layers. The change in the clay layers’ geometry, due to intercalation, involves modification in interlayer spacing, variation in the layers stacking mode, and diminishing of electrostatic forces between clay layers, resulting in great enhancement of the mechanical and thermal properties. The full separation of clay gallery due to polyaniline chains is called exfoliated structure, which is the third type of PANC nanocomposite [4,6]. As noted in the figures, separation of clay layers and formation of PANI inside the clay gallery were achieved. The characteristics of SCCO_2,_ such as its ability to swell and zero surface tension, selectivity facilitated the penetration of monomers inside the clay layers, to be oxidized by KPS to form PANI chains on and within clay layers [5]. The TEM image (Figure 2D) clarifies that the PANI formed within the clay layer using SCCO_2_ (10 MPa) was a nanorod structure, with an average diameter about 60 nm.

#### 3.1.2. Adsorbents BET Surface Area

Table 1 shows the BET surface area, total pore volume, and average pore size of prepared nanostructured PANI and PANC nanocomposite. It was noticed that both the average pore size and surface area of PANI were greater than the synthesized PANC nanocomposites. The decline in surface area may be referred to, as clay may block the active sites of PANI.

In previous studies, PANI-specific surface area was varied, depending on the synthesis technique or even the treatment of polyaniline after preparation. It was reported that the BET surface area ranged from 20.2 m^2^/g to 80 m^2^/g [6]. Hailing Xu et al. stated that the total pore volume and the average pore size varied dramatically after PANI treatment with chloroform, alongside the mean pore diameter increasing from 2.8 to 8.3 nm and the pore volume improving from 0.2 to 0.6 cm^3^/g, respectively [4]. In the current study, however, the pore volume is about 1.336 cm^3^/g, but the samples have a greater average pore size (35 nm), which may enhance the gas adsorption characteristics of the synthesized PANIs. Regarding the PANI nanocomposite surface area, it differed considerably when changing the material used in the nanocomposite’s preparation. Despite the high surface area of the PANI nanocomposites prepared in previous work, the total pore volume is nearly the same as the synthesized PANC. It was noticed that the prepared PANC had a higher average pore diameter (30 nm) than the previously prepared PANI nanocomposites [5].

#### 3.1.3. FTIR and XRD

The FTIR spectra of the synthesized PANI and PANC nanocomposite are shown in Figure 3. The samples were run in the wavelength range of 4000–400 cm^−1^. The characteristic PANI absorption bands occur at 1576.6, 1479.3, 1305.7, 1136.4, 3462.1, and 808.3 cm^−1^. These peaks are due to the Quinoid, Benzenoid rings, C–N stretching, C–H in-plane bending vibrations, N-H stretching vibrations, and C–H out-of-plane bending vibrations [29]. The synthesized PANI peaks are 1647.1, 1475.4, 1299, 1124.4, 3465.8, and 804.2, respectively. It was observed that there are minor differences between FTIR spectra of the prepared PANI and the PANI nanorods synthesized by Haibing et al., but they were close, with minimal difference due to the changes in preparation technique. There is noticeable shifting in 1589.2 cm^−1^ band down to 1576.6 cm^−1^, assigned to high doping level of the prepared PANI [26]. While comparing the PANI and PANC characteristic peaks, it was noticed that the FTIR spectrum of PANC showed no specific change or shift in the PANI peaks in the spectrum, except for decreasing the intensity of the N-H stretching vibrations, indicating that the incorporation of clay does not affect the chemical structure of PANI. This agreed with the research paper published by Yanrong Zhu et al. in 2022. However, Claudia María reported that adding clay altered the polyaniline’s FTIR spectrum of the resulted composite.

Figure 4 shows the XRD patterns of the synthesized PANI and PANC. The XRD spectra of the PANI and PANC show two characteristic peaks at 2θ: 20.1°, 26.2°, and 20.9°, 25.22° respectively. Moreover, there is a weak signal around 6.44°, which is related to pure clay interlayer spacing. It was observed that the peak intensity in the case of PANI was more intense than the PANC peaks, indicating that the prepared PANI sample is highly doped when compared to the PANC [3]. These results are consistent with the observations reported by Yoshimoto et al. and Ragupathy et al. [30,31].

The FTIR and XRD of PANI and PANC were obtained after adsorption, but the patterns were not significantly different from that obtained before adsorption, indicating that the chemical and the crystalline structure of PANI or PANC are not significantly changed by NH_3_ adsorption of the dye. This may be because the adsorption of ammonia on polyaniline was mainly physical adsorption, and the exposure of adsorbents samples to air after adsorption may release the adsorbed NH_3_ molecules [31].

### 3.2. Gas Adsorption Experiments

PANI and PANC were tested in relation to NH_3_ adsorption. For NH_3_ adsorption experiments, different inlet NH_3_ concentrations were prepared at a range of 35–150 ppm and then entered the fixed bed filter at a flow rate of 1 L/min and a fixed bed weight of 0.5 g adsorbent. Figure 5 illustrates the breakthrough curve for NH_3_ adsorption at different inlet concentrations. The removal of NH_3_ was at its maximum, at the start of the adsorption experiment. Then, after the breakthrough phenomena occurred, the outlet adsorbate concentration was elevated gradually with the increased time, until C_o_/C_i_ reached 0.9. The bed became saturated after around 35 min for the three studied inlet concentrations in the case of PANI; for PANC, the adsorption time increased and was reached at about 1 h. Figure 5. and 5. B show breakthrough curves for NH_3_ removal at different inlet gas concentrations (35–350 ppm) over PANI and PANC, respectively. Regarding the adsorption capacity values, there were significant rises in adsorption capacity as the inlet NH_3_ concentration increased (Table 2).

The interaction between PANI or PANC with NH_3_ is clear in the schematic diagram (Figure 6). Emeraldine salt (PANI or PANC) were deprotonated in the presence of NH_3_. As the gas concentration increased, the protonation level increased and formed energetically more favorable ammonium ion (NH_4_^+^). In addition, the emeraldine salt turned into the emeraldine base form [32]. Moreover, the high surface area and interconnected nanostructure of PANI- based adsorbent enhanced the diffusion of NH_3_ to react with more active sites of adsorbents molecules between the adsorbent and adsorbate [33]. The reason that PANC had better adsorption performance than PANI may be due to the complex structure of PANC and the increased contact time between the adsorbent and target gas. This is as a result of intercalation and exfoliation of PANC structure, which forces the gas to diffuse and come into contact with more PANC active sites. This led to the enhancement of NH_3_ adsorption.

Reviewing the adsorption capacities of different adsorbents toward NH_3_, PANI and PANC have higher adsorption capacities compared with other adsorbents tested in previous work. Yu Zhou et al. reported that α-MnO_2_ has the best adsorption capacity, reaching 17.76 mg NH_3_/g [34]. Xiaoxin Tian et al. stated that the adsorption performance was 26.5 mmol g^−1^, obtained at 25 °C and 1 bar [35]. The adsorption capacity of activated carbon towards NH_3_ gas reached 1.8 mg NH_3_/g carbon, according to Christiano C. Rodrigues et al. [20]. Other carbon-based materials have been extensively studied in the literature, but their adsorption capacities were lower that the adsorbents synthesized in this study. Bamboo charcoal, coconut shell activated carbon and coal-based activated carbon recorded 0.47 mg NH_3_/g, 0.35 mg NH_3_/g and 0.66 mg NH_3_/g adsorbent, respectively [34]. PANI and its nanocomposites were intensively examined in the literature as a gas sensor. While the studies investigated the use of PANI and its nanocomposite in gas adsorption, applications were limited. Jinwei Zhu used polyaniline/TiO_2_ composite in formaldehyde adsorption, and it showed an adsorption capacity of 0.67 mg g^−1^ [36]. Another study stated that PANI-derived carbon has been utilized in CO_2_ adsorption under low pressure, with an adsorption capacity of 1.0 mmol/g (at 0.15 atm) [37]. P. Tamilarasan et al. tested PANI/magnetite nanocapsules in CO_2_ capture at high pressures, and the adsorption capacity reached 47.5 mmol/g with 12 bar pressure at 28 °C [38]. Yuanzhen Chen et al. prepared a porous carbon from ferrocene-loaded polyaniline and utilized it in hydrogen adsorption; the hydrogen adsorption increased from 5.3 to 6.2 wt% at 77 K/5 MPa and 0.6 wt% to 0.85 wt% at 293 K/8 MPa [39]. Cristina Della Pina et al. synthesized PANI-based sorbents, which used the substitution of the toxic CS_2_ with the less hazardous CH_3_OH as the VOCs extraction solvent [40].

## 4. Conclusions

In this study, PANI and PANC were synthesized using the SC-CO_2_ assisted polymerization technique. The morphology of the prepared nanostructured materials was characterized using SEM and TEM techniques, yielding nanorod-like structures with an average diameter of about 80 nm in the case of PANI. TEM images revealed full separation of clay gallery within polyaniline chains, with an average diameter of about 60 nm. Moreover, both the surface area and mean pore diameter was decreased as a result of preparing the nanocomposite for PANI (24.6 m^2^/g, 35 nm) and PANC (22.9 m^2^/g, 30 nm) respectively, while the total pore volume increased as a result of adding clay. The synthesized nanostructured materials were tested for their ability to remove gaseous pollutants from the air (NH_3_). The obtained adsorption capacities were impressive at the studied concentration. The results revealed that the adsorption capacities of PANC nanocomposites were higher than nanostructured PANI. The adsorption capacities were 61.34 mgNH_3_/gm for PANC and 73.63 mgNH_3_/gm for PANI at the same inlet concentration (35 ppm). Therefore, PANI and PANC are capable of removing harmful pollutants from polluted air streams.

## Figures and Tables

**Figure 1 polymers-14-04533-f001:**
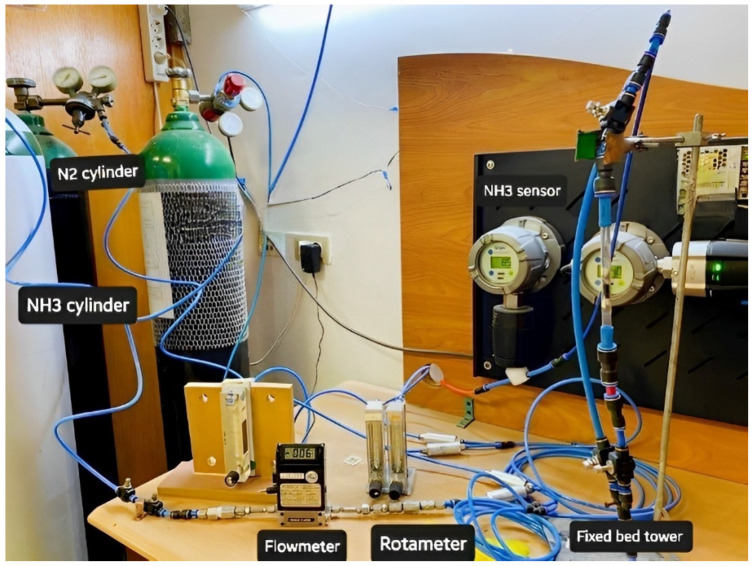
Experimental setup of ammonia preparation and adsorption system.

**Figure 2 polymers-14-04533-f002:**
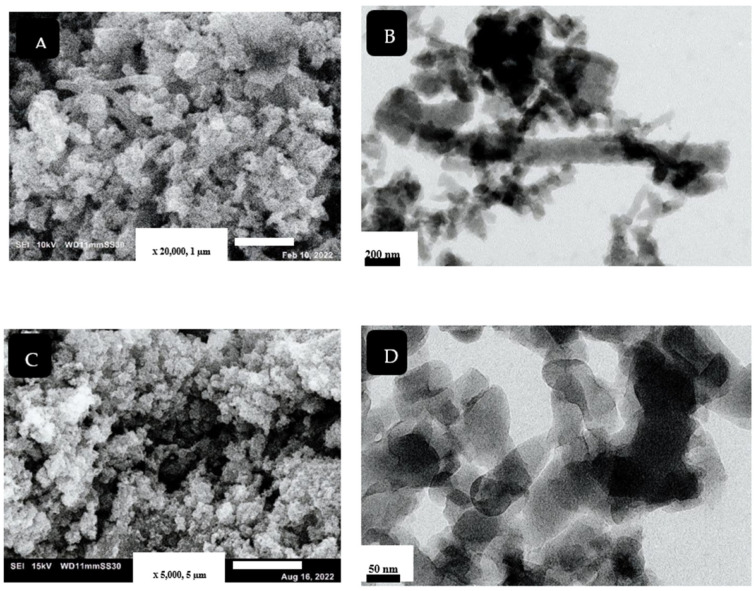
SEM and TEM images for PANI (**A**,**B**) and PANC (**C**,**D**), respectively.

**Figure 3 polymers-14-04533-f003:**
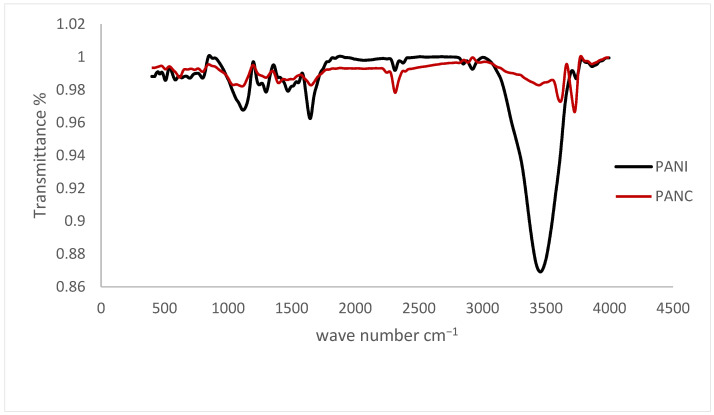
FTIR spectra for the PANI and PANC nanocomposite.

**Figure 4 polymers-14-04533-f004:**
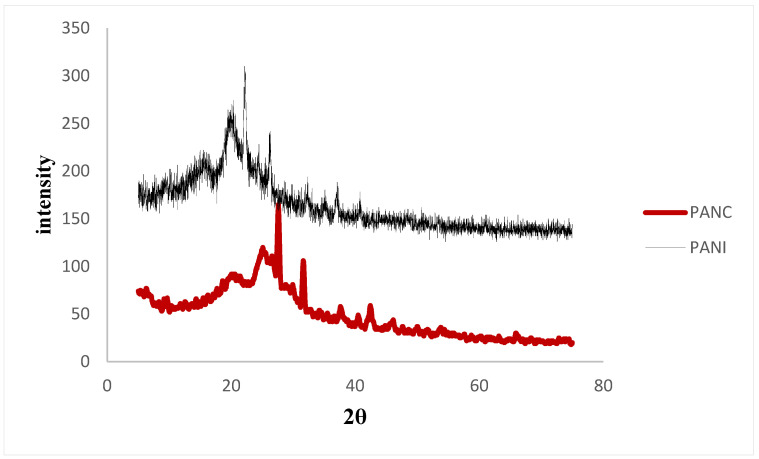
XRD for the PANI and PANC nanocomposite.

**Figure 5 polymers-14-04533-f005:**
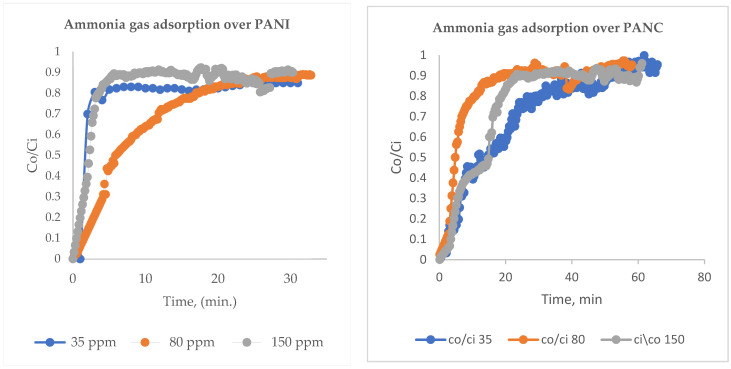
Breakthrough curves for NH_3_ removal over PANI and PANC at different inlet gas concentrations (35–150 ppm).

**Figure 6 polymers-14-04533-f006:**
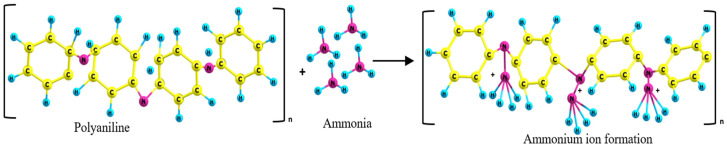
Schematic diagram of polyaniline-ammonia interaction.

**Table 1 polymers-14-04533-t001:** Surface area of the PANI and PANC.

Nanomaterials	Surface Area (m^2^/g)	Total Pore Volume (cm^3^/g)	Average Pore Size (nm)
PANI	24.61	0.1336	35
PANC	22.9	0.1753	30

**Table 2 polymers-14-04533-t002:** Adsorption capacity of NH_3_ over PANI and PANC at different inlet gas concentrations.

Concentration, (ppm)	Adsorption Capacity (mg/g PANI)	Adsorption Capacity (mg/g PANC)
35	61.34	73.63
80	333.2	400.16
150	485.3	582.4

## Data Availability

Not applicable.

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
