# Peer review of "Investigating the Adsorption Behavior of Polyaniline and Its Clay Nanocomposite towards Ammonia Gas"

_polymers, 2022, doi:10.3390/polym14214533_

Round 1

Reviewer 1 Report

The present manuscript entitled “Investigating the adsorption behavior of polyaniline and its clay nanocomposite towards ammonia gas” describes the aniline was polymerized without and with clay using carbon dioxide (CO2) assisted polymerization technique yielding different PANI and PANC samples, respectively. Furthermore, samples were characterized using different characterization methods such as XRD, FT-IR, TEM, SEM, and BET, etc., This present study revelated the impressive adsorption behavior of the prepared PANI and PANC nanomaterials towards NH3 gas. The manuscript is well organized, and the study is sufficiently performed. The result analysis is very accurate and adequate and lacks major errors. Therefore, I would recommend the publication of the manuscript in the Polymers Journal after some MINOR improvements.

I advise the authors to consider the following points while revising their manuscript.

·         Authors must check typos, use of is, are, was, were, and prepositions.

·         In the whole manuscript, the authors must be taken care of the superscripts and subscripts.

·         Abstract needs to be improved.

·         Some relevant references in this area are still missing in the introduction section, so include some significant relevant references from recent years to strengthen the introduction section.

·         Include the graphical abstract for the manuscript.

·         All figure's resolution is very poor, so provide high-resolution images.

·         In Figure 2, the SEM and TEM image scale bar is not properly visible, so draw the scale bar manually.

·         In FT-IR and XRD results: The authors should explore and discuss better their results with some more references in order to prepare a better discussion.

·         Check the reference style and maintain the journal names as abbreviations according to the MDPI format.

Reviewer 2 Report

In this work Amira et al., studied the adsorption behavior of polyaniline and its clay nanocomposite towards ammonia gas. PANI and PANC were characterized using different characterization methods. The synthesized nanomaterials were utilized as an adsorbent in a fixed bed configuration to investigate their adsorption behavior towards NH3. Three inlet NH3 concentrations were experimented, (35- 80 – 150 ppm). The results revealed that the adsorption capacities of PANC nano-composites were higher than nanostructured PANI for the studied concentrations. The adsorption capacities were 61.34 mgNH3/ gm PANC and 73.63 mgNH3/ gm PANI at the same inlet concentration (35 ppm). The work is well presented; however, some important investigations and discussions are missing, which are highlighted below.

As the work is mainly focused on ammonia adsorption using PANI and PANC composites, therefore the authors should discuss the applications of ammonia and the risk associated with ammonia by adding some relevant papers (Talanta 204 (2019): 713-730, Journal of Materials Chemistry C 10.4 (2022): 1326-1333 etc) to broaden the introduction of the paper.

The FTIR and XRD studies should be done after the adsorption of ammonia, so that the adsorption affect can be seen more clearly.

The SEM images are quite blurry, the quality of these images can be improved.

Ammonia adsorption mechanism can be schematically explained also.

The authors should highlight contact time also, as it does affect the adsorption.

The authors can perform electrochemical studies of the prepared composite before and after adsorption.

The authors should investigate adsorption studies of the prepared composites towards other gases (acetone, hydrogen, methane VOCs etc) also.

Round 2

Reviewer 2 Report

No more comments